# Bilateral Choanal Atresia and Endoscopic Surgery: A Chance for CHARGE Patients

**DOI:** 10.3390/jcm10132951

**Published:** 2021-06-30

**Authors:** Maria Baldovin, Diego Cazzador, Claudia Zanotti, Giuliana Frasson, Athanasios Saratziotis, Fabio Pagella, Stefano Pelucchi, Enzo Emanuelli

**Affiliations:** 1Department of Neurosciences, Section of Otorhinolaryngology, University of Padova, 35128 Padova, Italy; diego.cazzador@unipd.it (D.C.); zanotti.claudia89@gmail.com (C.Z.); enzoemanuelli@libero.it (E.E.); 2Unit of Otorhinolaryngology, Ospedale di Cittadella, 35013 Cittadella, Italy; giuliana.frasson@hotmail.it; 3Department of Otolaryngology, Head and Neck Surgery, University Hospital of Larissa, 41110 Larissa, Greece; asaratziotis@gmail.com; 4ENT Department, I.R.C.C.S. Policlinico San Matteo-University of Pavia, 27100 Pavia, Italy; tpagella@libero.it; 5ENT & Audiology Unit, Department of Neuroscience and Rehabilitation, University Hospital of Ferrara, 44124 Ferrara, Italy; stefano.pelucchi@unife.it

**Keywords:** choanal atresia, bilateral, CHARGE association, endoscopic surgery, stent, postoperative outcomes, restenosis, neonatal and pediatric airway disorders, nasal obstruction, congenital malformations

## Abstract

Bilateral choanal atresia (CA) is a rare congenital malformation frequently associated with other anomalies. CHARGE association is closely linked to bilateral CA. The aim of this study was to describe the outcomes of the endoscopic repair in bilateral CA, and to assess the role of postoperative nasal stenting in two cohorts of CHARGE-associated and non-syndromic CA. Thirty-nine children were retrospectively analyzed (16 patients had CHARGE-associated CA). The rate of postoperative neochoanal restenosis was 31.3% in the CHARGE population, and 47.8% in the non-syndromic CA cohort. Data on postoperative synechiae and granulation tissue formation, need for endonasal toilette and dilation procedures, and number of procedures per patient were presented. Stent positioning led to a higher number of postoperative dilation procedures per patient in the non-syndromic cohort (*p* = 0.018), and to a higher rate of restenosis both in the CHARGE-associated, and non-syndromic CA populations. Children with CHARGE-associated and non-syndromic bilateral CA benefitted from endonasal endoscopic CA correction. The postoperative application of an endonasal stent should be carefully evaluated.

## 1. Introduction

In 1979, Hall described 17 children with choanal atresia (CA) and multiple congenital anomalies [1], and Hittner reported on 10 children with ocular colobomas and multiple congenital anomalies, including CA. This association was called “Hall–Hittner syndrome” [2]. In 1981, Pagon et al. coined the acronym CHARGE to describe patients with coloboma, heart defect, CA, retarded growth and mental development, genital hypoplasia, and ear anomalies/deafness [3]. Because of the heterogeneous clinical presentation, in 1998, Blake et al. described four major criteria (the classical 4C’s: Choanal atresia, Coloboma, Characteristic ears, and Cranial nerve anomalies), and minor characteristics (genital hypoplasia, developmental delay, cardiovascular malformations, growth deficiency, orofacial cleft, tracheoesophageal fistula, and distinctive face) to define the new entity. Recently, some authors have proposed the inclusion of pathogenic CHD7 variant status as a major criterion [4]. Even if CHD7 mutation-positive and -negative patients do not differ in their chance of presenting CA [5], this condition has been reported in between 36% and 65% of cases [6,7], and frequently CA is bilateral. Otolaryngologists are involved in the treatment of several conditions in CHARGE patients, including hearing loss, inner ear malformations, and CA [8,9,10,11].

The endoscopic endonasal approach currently represents a widely accepted treatment modality for CA repair [12]. However, the use of a stent as an ancillary procedure after surgical CA correction is still debated. Literature regarding CA in CHARGE patients is scarce [13,14,15,16,17,18]. Up to now, no studies have compared stenting and non-stenting strategies for the surgical treatment of CA in the CHARGE population.

The aim of this study was to describe the outcomes of endoscopic CA repair in a population of patients with CHARGE association and in non-syndromic patients diagnosed with bilateral CA, as well as to assess the role of nasal stenting.

## 2. Materials and Methods

### 2.1. Selection of Patients

Between January 2001 and January 2016, 74 children diagnosed with congenital CA were managed at the Unit of Otolaryngology of the University of Padova. One patient was lost to follow-up, and in six cases details about first surgery were not available, resulting in seven patients excluded from the study (Figure 1).

This retrospective study included 39 children, 16 (41.0%) with diagnosis of CHARGE-associated bilateral CA and 23 (59.0%) with non-syndromic bilateral CA. Seventeen patients (43.6%) came to our institution for relapse of a previous surgical repair performed elsewhere.

Data were examined in accordance with Italian privacy and sensitive laws (D. Lgs. 196/03). Before surgery, all patients’ parents signed a detailed informed consent form and gave their written permission for clinical case publication. The diagnostic work-up included rigid/flexible video-rhino-laryngoscopy and computed tomography (CT) of the head.

### 2.2. Surgical Endoscopic Technique

Surgical procedures were performed under general anesthesia through an endoscopic approach in all cases, with 0° endoscopes (2.7 or 4 mm) connected to a high-definition camera and a monitor. Mucosal decongestion was induced by applying cotton pledges soaked in 5 mL of xylometazoline and 2 mL of lidocaine in 10 mL of saline solution with or without 1:100.000 epinephrine solution. Hegar’s dilators were used to puncture the atretic plate. The technique of mucoperiosteal flaps, shown in Figure 2, consisted of a star-shaped incision of the nasal mucosa at the junction between the vomer and the atretic plate to elevate the mucoperiosteal flap. The inferomedial portion of the atretic plate was drilled out with a skeeter, and the infero-posterior part of the vomer was resected with backbiting forceps. The anterior mucoperiosteal flap was used to cover the raw bony areas on the lateral wall of the nasal fossa and pterygoid plates. The mucosal flap of the nasopharyngeal side of the atretic plate resurfaced the medial side of the neochoana and the septum.

#### Stent

After surgical atresia correction, the use of stents was not routinely performed. In the case of choanal stenting, “U”-shaped endotracheal tubes 3.0–3.5 (Portex Ltd., Kent, UK) were employed and positioned through the neochoana with an anterior fixation at the columella. In the nasopharyngeal portion of the stent, three to four holes were created to allow breathing and postoperative irrigation.

### 2.3. Follow-Up and Postoperative Outcomes

Follow-up consisted of endonasal medications under sedation and local anesthesia, performed at days 7 and 14 after surgery. When no further medications under sedation were needed, office endoscopic evaluations were indicated at 45 and 90 postoperative days, then after six months and yearly.

The postoperative outcomes were the rate of restenosis, synechiae, and granulation tissue formation and the rate of toilettes and dilations procedures performed after surgery. The neochoana was considered restenosed in the case of total occlusion. Patency was defined if the neochoana lumen was wider than 50%. The toilette procedure consisted of removing debris and fibrinous tissue through aspiration or Weil forceps. For younger patients, this was performed under general anesthesia, while for collaborative children, performed under local anesthesia. The median follow-up was 26.5 months (IQR 12.5–50.5).

### 2.4. Analysis

Fisher’s exact test was used for comparison of categorical variables, while Mann-Whitney *U*-test was applied to compare continuous variables. Statistical significance was set at *p* < 0.05. Analyses were performed with SPSS, version 20 (Statistical Package for the Scientific Sciences, SPSS Inc., Chicago, IL, USA).

## 3. Results

All 39 patients included in the study were diagnosed with bilateral CA. The patients’ characteristics and data regarding stent positioning are presented in Table 1. The surgical outcomes adjusted for CHARGE association are summarized in Table 2. In the CHARGE population, 31.3% patients developed neochoanal restenosis, 18.8% had synechiae, and 31.3% granulation tissue development. The restenosis rate in non-syndromic-associated CA was 47.8%, while synechiae and granulation tissue were diagnosed in 26.1% and 60.9% of the patients, respectively. No significant differences were observed between the two study populations, nor in terms of postoperative outcomes, neither concerning the rate or the number of postoperative procedures per patient.

### 3.1. Role of a Stent in CHARGE-Associated Choanal Atresia

The surgical outcomes adjusted for postoperative stenting are summarized in Table 3. Overall, 31.3% patients developed a neochoanal restenosis, 18.8% synechiae, and 31.3% granulation tissue formation. No significant differences were calculated considering the rate of restenosis and the need for postoperative toilette and dilation procedures between the stented and non-stented patients (40.0% vs. 16.7%, *p* = 0.588; 30.0% vs. 33.3%, *p* = 1.00; 50.0% vs. 66.7%, *p* = 0.633, respectively).

Although not statistically significant, the mean number of toilette procedures (0.40 ± 0.70 vs. 1.67 ± 3.61) and dilations (1.40 ± 1.65 vs. 1.67 ± 2.25) per patient was lower in the group with stent.

### 3.2. Role of a Stent in Non-Syndromic Choanal Atresia

The surgical outcomes adjusted for postoperative stenting are summarized in Table 4. The rate of restenosis and granulation tissue formation was higher for the group with a stent placement than for patients without a stent (63.6% vs. 33.3%, *p* = 0.220 and 81.8% vs. 41.7%, *p* = 0.089). Patients who underwent stent positioning required toilette and dilation procedures in 54.5% and 90.9% of cases, respectively. Although statistical significance was not reached, lower rates were observed for the patients without a stent (33.3% and 58.3% of toilette and dilation procedures, respectively).

Interestingly, stented patients needed a significantly higher number of dilations per patient, than non-stented patients (2.0 (2–4) vs. 1.0 (0–2), *p* = 0.018]).

## 4. Discussion

The results obtained from this study demonstrated that the endoscopic technique is a valuable option for children with bilateral CA. The first endoscopic approach to treat CA was described by Stankiewicz in 1990 [19], who reported on four cases with detailed explanation of the surgical technique and the reasons for success and failure. In 2000, an international survey of pediatric otolaryngologists belonging to the American Society of Pediatric Otolaryngology (ASPO) stated that 85% of the interviewees preferred the endoscopic approach for CA repair, even if transpalatal repair and puncture with Fearon dilators were still advocated by 60% and 17%, respectively. This survey underlined controversies in the management of CA among the experts [20]. Approximately 20 years later, Moreddu et al. submitted to the members of the International Otolaryngology Group (IPOG) a questionnaire to establish expert recommendations on CA management and care [12]. Transnasal endoscopic repair was the preferred initial approach for CA repair, with 35.7% of the members adopting the mucosal flap technique. The transpalatal approach was applied when transnasal repair was prevented. Progressively, the use of a stent found less indication, and only in selected cases.

In non-syndromic bilateral CA, we obtained 66.7% vs. 36.4% success rates (*p* = 0.220) for the stented and non-stented groups, respectively. Compared to stented patients, a lower rate of non-stented patients underwent endonasal toilette procedures (33.3% vs. 54.5%, *p* = 0.414) and dilations (58.3 vs. 90.9%, *p* = 0.155), with a significantly lower mean number of dilations per patient (1.33, vs. 2.91, *p* = 0.018)

The low incidence of CA prevents designing a high level of evidence-based studies to address open issues. Systematic reviews with meta-analyses have thus been conducted, trying to clarify the debated use of a nasal stent after CA repair. Bedwell and Choi examined five studies, including 112 patients, with the aim of comparing outcomes between stented and non-stented patients. They affirmed that stent placement is not necessary after endoscopic surgical repair to obtain excellent postoperative outcomes and low complication rates [21]. Strychowsky’s meta-analysis of 12 studies, including 215 patients, revealed no differences in terms of success rate between patients with and without a stent [22]. The only prospective randomized controlled study was designed by Tomoum et al. [23], which divided 72 patients into two groups according to the use of a stent. The postoperative outcomes were significantly better in the group without a stent.

This study included a cohort of 16 patients diagnosed with CHARGE association, all presenting with bilateral CA. The endoscopic approach proved to be an effective technique for the treatment of associated CA, ensuring in CHARGE patients a 68.7% postoperative success rate and a low incidence of synechiae and granulation tissue formation (31.3%).

In literature, there are very few studies describing the outcomes of endoscopic CA repair in CHARGE association (Table 5). The results are hardly comparable, given the several surgical approaches included in the series [13,14,24]. More recent studies have described cases treated only with the endoscopic technique [15,16,17,18].

Schraff et al. first compared postoperative outcomes in 14 CHARGE patients (nine with bilateral CA and five with unilateral CA), treated via transnasal (10 cases) or transpalatal (four cases) approaches. The authors supported the primary transpalatal approach for patients with bilateral atresia [13]. Hengerer et al. included in their analysis 16 patients treated with transpalatal (11 procedures), transnasal (seven procedures), and endoscopic approaches (seven procedures). They confirmed the superiority of the endoscopic technique to reduce the risk of restenosis in CHARGE children [14]. The largest sample of CHARGE patients (20 cases) was collected in a study by Moreddu et al. [24]. They analyzed patients with both bilateral and unilateral CA (114 cases) treated surgically with different techniques (transpalatal and endoscopic with or without stent placement) at a single institution during 30 years of experience. Unfortunately, the authors did not differentiate outcomes adjusting for the diagnosis of CHARGE association. Interestingly, however, they observed a correlation between CHARGE association and an increased number of surgical interventions (2.85 vs. 2.16; *p* = 0.02) [21]. Sinha et al. used Hegar’s dilators and nasal stents in 22 cases. In their experience, CHARGE association had a very unsolicited outcome. All CHARGE patients (eight cases) died 5–10 days after surgery due to complications of the syndrome [15]. A recent Italian multicentric study included 84 patients with CA. The endoscopic technique was applied for all patients and a stent was placed at the beginning of the experience in some unilateral CA cases [16]. Sixteen were affected by CHARGE association (10 bilaterally and six unilaterally), and two of them (both bilateral) required revision surgery. Gulsen et al. collected six CHARGE children without mention of outcomes. Intriguingly, the authors concluded that the presence of congenital malformation associated with the atresia is one of the negative predictors for the successful rate of endoscopic repair [17]. The study of Brihaye showed good postoperative outcomes for the four CHARGE patients that demonstrated healing type I (normal healing) or II (limited scar formation and no breathing impairment) during follow-up [18].

The success rate adjusted for the use of a stent obtained in this study in CHARGE-associated CA was higher for non-stented patients (83.3%) than stented patients (60.0%), even if the difference was not statistically significant. Analogously to what was observed in the population with non-syndromic CA, also in the CHARGE population, patients treated without a stent needed a higher mean number of postoperative endonasal toilette procedures per patient, thus underlining the importance of adequate postoperative care to avoid concentric fibrosis and restenosis when a stent is not indicated.

There are several weaknesses in this study. The retrospective nature of the study did not allow us to standardize treatments and postoperative procedures. Prospective studies should implement the level of evidence on this topic, but the extremely low incidence rate of this clinical represents a limitation. Moreover, this study presented a lack of information on the prognostic role of the different surgical endoscopic methods applied for CA repair, which was beyond the scope of the paper. Further research should investigate the efficacy of the mucoperiosteal flap technique in surgical CA correction. Although the study population was too small to make definitive conclusions, the main strengths of the present study lie in the homogeneous series of consecutive patients included, and —probably for the first time—in the investigation on the role of stent in postoperative outcomes for CHARGE patients.

## 5. Conclusions

Children with CHARGE association and bilateral CA benefit from endonasal endoscopic correction of the atretic plate, showing comparable results to that observed for non-syndromic-associated bilateral CA. Endonasal stent positioning led to the need for a significantly higher number of postoperative dilation procedures per patient in the non-syndromic cohort. Although not statistically significant, data regarding stent application both in CHARGE and non-syndromic children revealed a higher rate of restenosis. Conversely, a higher number of endonasal toilette procedures per patient was registered in non-stented patients, thus underlying the need for meticulous postoperative care when endoscopic stent-free CA correction is preferred.

## Figures and Tables

**Figure 1 jcm-10-02951-f001:**
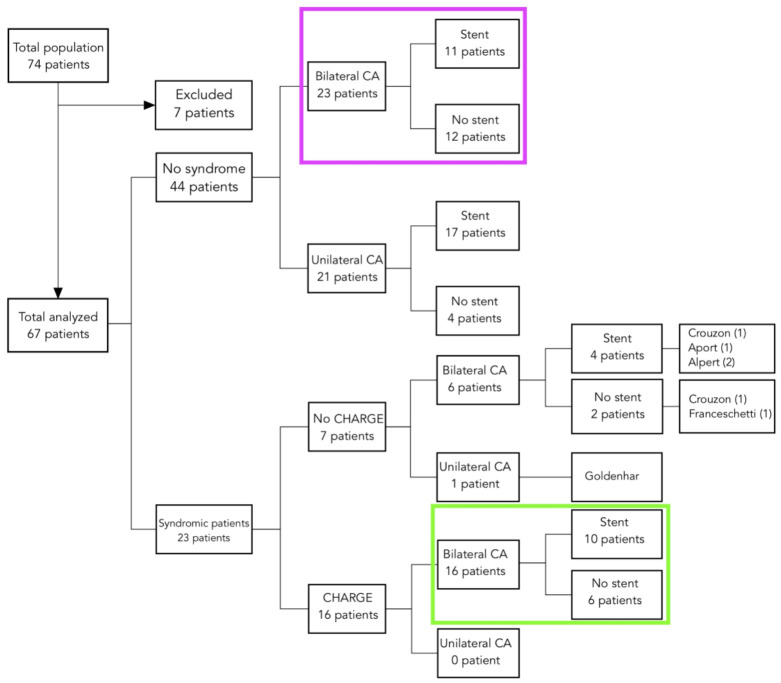
Diagram showing the overall population divided according to the association with syndromic conditions, laterality of the atresia, and stenting. Colored squares indicate the study populations. CA, choanal atresia.

**Figure 2 jcm-10-02951-f002:**
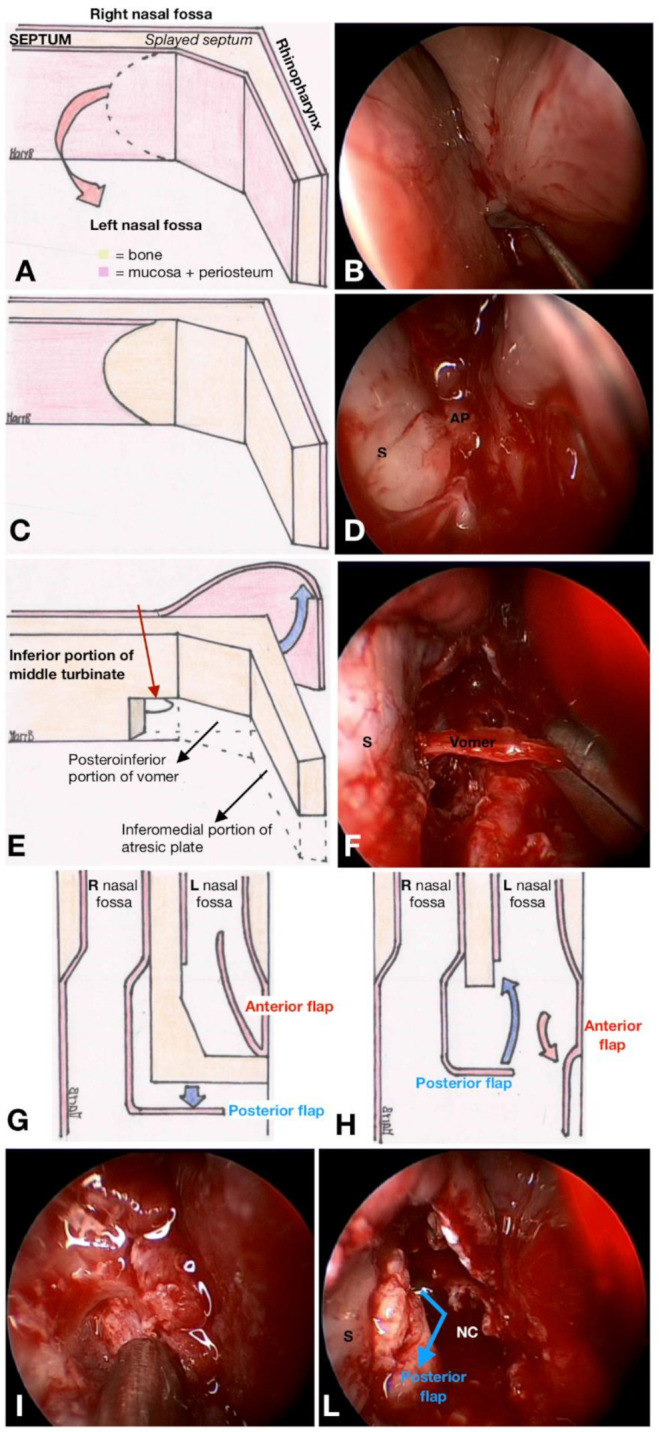
**The mucoperiosteal flap technique.** (**A**) Drawing of a left choanal atresia (CA). Dashed line indicates the incision made for harvesting the anterior mucoperiosteal flap, to be positioned to cover the lateral wall of the nasal cavity thereafter. An equivalent endoscopic image is shown in (**B**). (**C**,**D**) The subperiosteal plane evidenced after mucoperiosteal flap removal. (**E**) Drawing of the posterior mucoperiosteal flap created on the nasopharyngeal side of the contralateral nasal fossa, flipped contralaterally (blue arrow). Dashed lines underline the bony portion of the atretic plate. The inferior portion of the middle turbinate of the contralateral nasal fossa represents the anterior landmark for the posterior margin of vomer resection (thin red arrow). (**F**) Endoscopic view of the vomer resection. (**G**,**H**) Repositioning of the harvested flaps in the neochoana. The anterior flap (both in cases of unilateral and bilateral CA) covers the lateral wall of the nasal fossa; the posterior flap (in unilateral CA) resurfaces the free margin of the vomer bone (L = left; R = right). (**I**,**L**) Equivalent endoscopic views. (**I**) Nasopharyngeal mucosal plane observed after removing the atretic plate. (**L**) After anterior replacement (blue arrow), the posterior mucoperiosteal flap harvested on the nasopharyngeal side of the CA covers the posterior edge of the vomer (S = septum; AP = atretic plate).

**Table 1 jcm-10-02951-t001:** Demographic characteristics and data of the study cohort.

	CHARGE-Associated CA	Non-Syndromic CA
	*n* = 16	*n* = 23
Age (days), median (IQR)	5 (3–83)	15 (2.5–165)
Sex		
Male	5 (31.3%)	10 (43.5%)
Female	11 (68.7%)	13 (56.5%)
Stent		
No	6 (37.5%)	11 (47.8%)
Yes	10 (62.5%)	12 (52.2%)
Stenting duration (days), median (IQR)	42 (41–46)	41 (31.5–42.5)

CA, choanal atresia; IQR, interquartile range.

**Table 2 jcm-10-02951-t002:** Postoperative outcome comparison between CHARGE-associated and non-syndromic bilateral choanal atresia.

	CHARGE-Associated CA	Non-Syndromic CA	*p*-Value
	*n* = 16	*n* = 23	
Restenosis, *n* (%)	5 (31.3%)	11 (47.8)	0.342
Synechiae, *n* (%)	3 (18.8%)	6 (26.1)	0.711
Granulation tissue, *n* (%)	5 (31.3%)	14 (60.9)	0.105
Toilettes, *n* (%)	5 (31.3%)	10 (43.5)	0.517
Dilations, *n* (%)	9 (56.3%)	17 (73.9)	0.312
N° of toilettes per patient			
Mean (SD)	0.93 (2.31)	0.78 (1.24)	
Median (IQR)	0 (0–1)	0 (0–1)	0.637
N° of dilations per patient			
Mean (SD)	1.60 (1.84)	2.09 (1.90)	
Median (IQR)	1 (0–3)	2 (0–3)	0.425

SD, standard deviation; IQR, interquartile range.

**Table 3 jcm-10-02951-t003:** Postoperative outcomes in CHARGE-associated choanal atresia according to stenting/non-stenting procedure.

CHARGE-Associated CA	Stent (S)	Non-Stent (NS)	S vs. NS
	*n* = 10	*n* = 6	*p*-Value
Restenosis, *n* (%)	4 (40.0)	1 (16.7)	0.588
Synechiae, *n* (%)	2 (20.0)	1 (16.7)	0.868
Granulation tissue, *n* (%)	3 (30.0)	2 (33.3)	1
Toilettes, *n* (%)	3 (30.0)	2 (33.3)	1
Dilations, *n* (%)	5 (50.0)	4 (66.7)	0.633
N° of toilettes per patient			
Mean (SD)	0.40 (0.70)	1.67 (3.61)	
Median (IQR)	0 (0–1)	0 (0–1)	0.955
N° of dilations per patient			
Mean (SD)	1.40 (1.65)	1.67 (2.25)	
Median (IQR)	0 (0–3)	1 (0–2)	1

SD, standard deviation; IQR, interquartile range.

**Table 4 jcm-10-02951-t004:** Postoperative outcomes in non-syndromic bilateral choanal atresia according to stenting/non-stenting procedure.

Non-Syndromic Bilateral CA	Stent (S)	Non-Stent (NS)	S vs. NS
	*n* = 11	*n* = 12	*p*-Value
Restenosis, *n* (%)	7 (63.6)	4 (33.3)	0.220
Synechiae, *n* (%)	1 (9.1)	5 (41.7)	0.155
Granulation tissue, *n* (%)	9 (81.8)	5 (41.7)	0.089
Toilettes, *n* (%)	6 (54.5)	4 (33.3)	0.414
Dilations, *n* (%)	10 (90.9)	7 (58.3)	0.155
N° of toilettes per patient			
Mean (SD)	0.64 (0.67)	0.92 (1.62)	
Median (IQR)	1.0 (0.0–1.0)	0.0 (0.0–1.5)	0.74
N° of dilations per patient			
Mean (SD)	2.91 (1.81)	1.33 (1.72)	
Median (IQR)	2.0 (2.0–4.0)	1.0 (0.0–2.0)	0.018 *

SD, standard deviation; IQR, interquartile range. * Statistical significance.

**Table 5 jcm-10-02951-t005:** Studies on choanal atresia repair in CHARGE association.

Author, Year	Patients *n*	Study Period (Years)	CHARGE *n*	Bilateral *n*	Endoscopic Approach	Outcome * *n*, (%)
Schraff et al., 2006 [13]	57	1990–2005	14	9	10	4 (40.0)
Hengerer et al., 2008 [14]	73	1973–2005	16	na	7	1 (14.3)
Sinha et al., 2016 [15]	22	20 years	8	na	8	100% mortality due to CHARGE-related conditions
Karligkiotis et al., 2017 [16]	84	1996–2013	16	10	16	2 (12.5)
Gulsen et al., 2017 [17]	48	2000–2014	6	na	6	na
Brihaye et al., 2019 [18]	36	1999–2015	4	na	4	healing type I or II
Moreddu et al., 2020 [24]	114	1986–2016	20	10	na	na
Our study, 2021	67	2001–2016	16	16	16	5 (31.3)

CA, choanal atresia; na, not available. * Revision surgery for restenosis after endoscopic CA repair.

## Data Availability

The data presented in this study are available on request from the corresponding author.

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
