# Peer review of "Bilateral Choanal Atresia and Endoscopic Surgery: A Chance for CHARGE Patients"

_jcm, 2021, doi:10.3390/jcm10132951_

Round 1
Reviewer 1 Report
The study “Choanal atresia and endoscopic surgery: a chance for CHARGE patients” covers an interesting topic regarding the usefulness of nasal stents in different cohorts of patients with choanal atresia. Considering the fact that it is an orphan disease, the number of 67 patients treated in one medical center is remarkable.
Prior to a possible publication there are several comments that should be considered:
- Surgical endoscopic techniques: 1 ml epinephrine 1:2000 and 10 ml lidocaine: Is there more experience with this mixture? Especially epinephrine seems to be applied in a high concentration in really young patients.
- The whole manuscript should be revised by a native speaker as there are many spelling mistakes throughout the text e.g. atresic instead of atretic (l. 104, inconsistent use), drowing instead of drawing (l. 109), controlateral instead of contralateral (l. 110), etc.
- The procedure “toilettes” should be explained in more detail. What is performed exactly and is it performed in general anaesthesia?
- l. 338: all CHARGE patients (8 cases) died 5–10 days after surgery due to complications of the syndrome. (no "th" after 5 and 10)
- l. 353: underwent toilettes (no “to”)
- l. 372: […] was lower in CHARGE than in NO CHARGE children.
- L. 375 “after” instead of “from”
Author Response
Reviewer 1 Comments:
● The study “Choanal atresia and endoscopic surgery: a chance for CHARGE patients” covers an interesting topic regarding the usefulness of nasal stents in different cohorts of patients with choanal atresia. Considering the fact that it is an orphan disease, the number of 67 patients treated in one medical center is
remarkable.
► Thank you for your positive comments. Sixty-seven patients covered the experience of our institution indeed, however in the revised version of the manuscript we decided to focus on 39 patients, thus including only children with bilateral choanal atresia (CA) (16 CHARGE and 23 with non-syndromic CA). The title has been modified accordingly: “Bilateral choanal atresia and endoscopic surgery: a chance for CHARGE patients”. The abstract and several parts of the manuscript has been modified accordingly, as well. Results now focuses on postoperative outcomes for CHARGE-associated CA, and non-syndromic CA. Then, results adjusted for the use of stent in the two study populations are presented.
● Surgical endoscopic techniques: 1 ml epinephrine 1:2000 and 10 ml lidocaine: Is there more experience with this mixture? Especially epinephrine seems to be applied in a high concentration in really youngpatients.
► That concentration were mistakenly reported. The sentence has been corrected: “Mucosal decongestion was induced applying cotton pledges soaked in 5 ml xylometazoline, 2 ml lidocaine in 10 ml saline solution with or without 1:100.000 epinephrine solution”. This mixture has been used in clinical practice since many years. For reference see: Emanuelli E, Bossolesi P, Borsetto D, D'Avella E. Endoscopic repair of cerebrospinal fluid leak in paediatric patients. Int J Pediatr Otorhinolaryngol. 2014;78(11):1898-1902. doi:10.1016/j.ijporl.2014.08.020.
● The whole manuscript should be revised by a native speaker as there are many spelling mistakes throughout the text e.g. atresic instead of atretic (l. 104, inconsistent use), drowing instead of drawing (l. 109), controlateral instead of contralateral (l. 110), etc.
► Thank you for your suggestions. The entire manuscript has been grammatically revised and typos corrected.
● The procedure “toilettes” should be explained in more detail. What is performed exactly and is it performed in general anaesthesia?
► Thank you for this remark. We added a more detailed explication in the Materials and Methods section, as follows: “Toilette procedure consisted in removing debris and fibrinous tissue through aspiration or Weil forceps. For younger patients it was performed under general anesthesia, while for collaborative children under local anesthesia”.
● 338: all CHARGE patients (8 cases) died 5–10 days after surgery due to complications of the syndrome. (no "th" after 5 and 10)
► Thank you for your observation, the sentence has been corrected.
● 353: underwent toilettes (no “to”)
372: [...] was lower in CHARGE than in NO CHARGE children.
375 “after” instead of “from”
► Than you for your corrections, the entire manuscript has been grammatically revised and typos corrected.
Reviewer 2 Report
This is an article that should describe the results of stenting (or not) in CHARGE patients, but:
- all patients are included
- many factors are studied.
That makes this article really difficult to read and understand.
The only significant result about the problematic exposed in the introduction is the presence of granulation tissue.
Some results are counter-intuitive: why bilateral CA have more dilatations when stenting is performed, but not in CHARGE patients ? Why non CHARGE patients have worse results than CHARGE ?
This leads to the main concern about this study: the methodology is not appropriate. It is not conducted using a multivariate analysis, whereas age at surgery, type of surgery (surgical technique), type of atresia, syndroms are critical confounding factors when faced with choanal atresia.
The conclusions of this article are not supported by the results.
Author Response
Reviewer 2 Comments:
● This is an article that should describe the results of stenting (or not) in CHARGE patients, but: all patients are included
many factors are studied.
That makes this article really difficult to read and understand.
► Thank you for your comments. Sixtyseven patients covered the experience of our institution indeed, however in this revised version of the manuscript we decided to focus on 39 patients, thus including only children with bilateral choanal atresia (CA) (16 CHARGE and 23 with non-syndromic CA). The title has been modified accordingly: “Bilateral choanal atresia and endoscopic surgery: a chance for CHARGE patients”. The abstract and several parts of the manuscript has been modified accordingly, as well. In addition, detailed description of the entire casuistic has been removed. Results now focuses on postoperative outcomes for CHARGE-associated CA, and non-syndromic CA. Then, results adjusted for the use of stent in the two study populations are presented.
● The only significant result about the problematic exposed in the introduction is the presence of granulation tissue.
► The introduction has been revised. Results now focuses on postoperative outcomes for CHARGE-associated CA, and non-syndromic CA. Then, results adjusted for the use of stent in the two study populations are presented.
● Some results are counter-intuitive: why bilateral CA have more dilatations when stenting is performed, but not in CHARGE patients ? Why non CHARGE patients have worse results than CHARGE ?
► Definitely, the study population is too small to make definitive conclusions, as we stated in the Discussion section of the manuscript (paragraph on strengths and limitations of the study). However, our findings are in line with other authors’ results. In the meta-analysis conducted by Strychowsky et al. in 2015, the need for revision surgery was not statistically significantly different between CHARGE children compared with non-CHARGE children.
● This leads to the main concern about this study: the methodology is not appropriate. It is not conducted using a multivariate analysis, whereas age at surgery, type of surgery (surgical technique), type of atresia, syndroms are critical confounding factors when faced with choanal atresia.
► Thank you for your observation. We added a detailed paragraph on study limitations at the end of the Discussion: “There are several weaknesses to this study. The retrospective nature of the study does not allow us to standardize treatments and postoperative procedures. Prospective studies would implement the level of evidence on this topic, but the extremely low incidence rate of this clinical represents a limitation. Moreover, this study presents a lack of information on the prognostic role of the different surgical endoscopic methods applied for CA repair, which was beyond the scope of the paper. Further research should investigate the efficacy of mucoperiosteal flap technique in surgical CA correction. Although the study population is too small to make definitive conclusions, the main strengths of the present study lie in the homogeneous series of consecutive patients included, and -probably for the first time- in the investigation on the role of stent in postoperative outcomes for CHARGE patients”.
● The conclusions of this article are not supported by the results.
► Conclusions has been reformulated in accordance to the revision of results: “Children with CHARGE association and bilateral CA benefit from endonasal endoscopic correction of the atretic plate, showing comparable results to that observed for non-syndromic associated bilateral CA. Endonasal stent positioning led to the need for a significantly higher number of postoperative dilations procedures per patients in the non-syndromic cohort. Although not statistically significant, data regarding stent application both in CHARGE and non-syndromic children revealed a higher number of restenosis rate. Conversely, a higher number of endonasal toilette procedures per patient was registered in non-stented patients, thus underlying the need for meticulous postoperative care when the endoscopic stent-free CA correction is preferred”.